# Management of Bleeding from Unresectable Gastric Cancer

**DOI:** 10.3390/biomedicines7030054

**Published:** 2019-07-24

**Authors:** Hideaki Kawabata, Misuzu Hitomi, Shigehiro Motoi

**Affiliations:** Department of Gastroenterology, Kyoto Okamoto Memorial Hospital, 100 Nishinokuchi, Sayama, Kumiyama-cho, Kuze-gun, Kyoto 613-0034, Japan

**Keywords:** gastric cancer, bleeding, endoscopy, transcatheter arterial embolization, palliative radiotherapy

## Abstract

Bleeding from unresectable gastric cancer (URGC) is not a rare complication. Two major ways in which the management of this issue differs from the management of benign lesions are the high rate of rebleeding after successful hemostasis and that not only endoscopic therapy (ET) and transcatheter arterial embolization (TAE) but palliative radiotherapy (PRT) can be applied in the clinical setting. However, there are no specific guidelines concerning the management of URGC with bleeding. We herein discuss strategies for managing bleeding from URGC. A high rate of initial hemostasis for active bleeding is expected when using various ET modalities properly. If ET fails in patients with hemostatic instability, emergent TAE is considered in order to avoid a life-threating condition due to massive bleeding. Early PRT, especially, regimens with a high biologically effective dose (BED) of ≥39 Gy should be considered not only for patients with hemostatic failure but also for those with successful hemostasis and inactive hemorrhage, as longer duration of response with few complications can be expected. Further prospective, comparative studies considering not only the hemostatic efficacy of these modalities but the patients’ quality of life are needed in order to establish treatment strategies for bleeding from URGC.

## 1. Introduction

Gastric cancer (GC) is the fifth most common cancer type worldwide, with an estimated 1.0 million new cases annually in 2018, and the third most common cause of cancer death, with an estimated 783,000 death annually [1]. Gastrointestinal (GI) malignancies, mainly GC, account for 2–8% of upper GI bleeding events [2]. GC often induces local symptoms, such as pain, gastric outlet obstruction, and bleeding from the primary site. In particular, bleeding from GC accounts for 58% of bleeding cases resulting from upper GI malignancies [3] and not only compromises the quality of life but also occasionally results in a life-threatening condition. Indeed, failed bleeding control is associated with a poor prognosis [4].

Radical gastrectomy is the ideal treatment option both for curing GC and relieving these local symptoms, and a favorable outcome has been reported in GC patients with overt bleeding [5]. However, for patients with unresectable factors, such as locally advance disease and metastasis, curative surgical resection is not applicable. In addition, patients with a poor general condition are generally not candidates for surgical treatment.

According to the European, Japanese and American Guidelines for the diagnosis and management of nonvariceal upper GI hemorrhage (NVUGIH), following hemodynamic resuscitation, early upper GI endoscopy and endoscopic treatment is recommended [6,7,8]. However, published data on the role of endoscopic hemostasis in bleeding due to upper GI tract neoplasia are limited, and evidence supporting a specific modality is scarce [6]. Furthermore, the Japan Gastroenterological Endoscopy Society Guideline for endoscopic management of non-variceal upper GI bleeding include no description of bleeding from GI malignancies [7].

The characteristics and management of GC bleeding and those of peptic ulcer bleeding share some common points. However, bleeding GC differs from bleeding peptic ulcers in terms of its gross appearance, behavior, and response to proton pump inhibitors (PPI). Therefore, treatment strategies in patients with unresectable GC bleeding should be discussed separately.

Several less-invasive treatment modalities for unresectable GC bleeding have been proposed, including endoscopic hemostasis, transcatheter embolotherapy, and radiotherapy (RT). We herein review the therapeutic roles of these modalities and discuss strategies for managing bleeding from unresectable GC. Although we will focus on these treatment modalities, it is also important to understand the importance of the initial patient evaluation, hemodynamic resuscitation, and pre-endoscopy management, including pharmacological therapy, as described in the guidelines [6,7,8].

## 2. Endoscopic Management for Bleeding from Unresectable GC

Literature searches were conducted in PubMed databases published in English for the period since 2013 using related key words, such as ‘GC’, ‘bleeding’ and ‘endoscopic’ (Figure 1). We found five recent studies reporting on the outcomes of endoscopic therapy (ET) for GC bleeding [4,9,10,11,12] (Table 1) and a review article [13]. Four of these five reports were retrospective studies using conventional hemostatic modalities [4,9,10,11], whereas one prospectively investigated the suitability of hemostatic powder application as an initial endoscopic treatment modality [12].

### 2.1. Outcomes of ET

The rate of successful hemostasis by ET in patients with GC bleeding was as high as 73–100%, except in one study with a low success rate of 31% [9]. This result compares favorably with the initial hemostasis rates in patients with peptic ulcer bleeding (approximately 90%) [7]. A high rate of sustained hemostasis was also achieved in case of bleeding from peptic ulcers, resulting in a rebleeding rate after initial ET of only 2–10% [7]. However, the rebleeding rates in patients who underwent ET for GC bleeding were approximately 30–40% [4,10,11] using conventional ET, which are obviously inferior to the rates in patients with peptic ulcer bleeding. Repeat ET was attempted for 11–78% of rebleeding patients [4,10,11], and hemostasis was achieved in 88.9% of them in one study [10]. However, another study reported seven out of eight patients with rebleeding or failed initial hemostasis died after repeat ET [4].

Previous studies reported a median overall survival (OS) and 30-day mortality rate of approximately 3–6 months and 0–22% after ET, respectively [4,9,10,11,12], although prognosis after ET depends on not only the efficacy of treatment but also tumor progression, the general condition, and concomitant therapy, such as chemotherapy and RT. Koh et al. [9] found that the 30-day mortality rates in patients with successful and failed endoscopic hemostasis were 0% and 25.8%, respectively. Furthermore, two detailed analyses showed that a high rebleeding was associated with a poor prognosis. The median OS after initial hemostasis was significantly lower in patients with rebleeding than in those without rebleeding (2.7 vs. 3.9 months, *p* = 0.020) [11]. In another study, the median OS in patients who experienced early rebleeding (≤3 days after initial hemostasis) was significantly lower than that of patients who experienced late bleeding (>3 days after initial hemostasis) or no rebleeding (1.0 months vs. 3.1 months vs. 4.3 months, respectively; *p* = 0.004) [10].

### 2.2. Indications of ET

The Forrest classification of endoscopic stigmata [14] is useful for planning treatment strategies for bleeding GC. The Forrest classification is defined as follows: Forrest Ia, spurting hemorrhage; Forrest Ib, oozing hemorrhage; Forrest IIa, nonbleeding visible vessel; Forrest IIb, an adherent clot; Forrest IIc, flat pigmented spot; and Forrest III, clean base ulcer.

Two studies described the indications of ET for GC bleeding, as Forrest Ia, Ib, IIa, and IIb [10,11]. The Forrest classification in most cases of bleeding GC was shown to be Forrest I (61–100%) [9,10,11,12]. Among them, Forrest Ib in particular accounted for a large portion of endoscopic findings of stigmata. Song et al. [11] showed that the distribution of the Forrest classification among patients undergoing ET for GC bleeding was 5% Forrest Ia, 55% Forrest Ib, 13% Forrest IIa, and 25% Forrest IIb. They also showed that the endoscopic active bleeding was not a predictive factor for rebleeding after initial hemostasis according to a univariate analysis.

### 2.3. Modalities of ET

The excellent outcomes in the high rate of initial hemostasis for GC bleeding are mainly attributed to the development and improvement of endoscopic hemostatic modalities including injection therapy (absolute ethanol, hypertonic saline epinephrine solution), mechanical therapy of hemoclip placement, and thermal therapy (hemostatic forceps, argon plasma coagulation [APC], heater probe thermocoagulation), as well as a combination of several modalities [6,7,13].

For patients with GC bleeding, thermal therapy, such as with hemostatic forceps or APC, was used most often to achieve initial hemostasis (79–93%) (Figure 2), while hemoclipping was performed in only 5–25% of patients. Combined hemostatic methods were required in 34–44% of patients [4,10,11].

Kim et al. [10,13] discussed the choice of hemostatic modalities should be based on the Forrest classification. For patients with spurting hemorrhage (Forrest Ia) and non-bleeding visible vessels (Forrest IIa), electrocoagulation using hemostatic forceps was commonly used and controlled the bleeding more effectively than hemoclips (88% vs. 70%). In contrast, for patients with oozing hemorrhage (Forrest Ib), electrocoagulation using APC was commonly applied and effectively controlled the bleeding. Performing electrocoagulation for oozing hemorrhage seems to be reasonable because APC can ablate the extensive, diffuse lesions often seen in this bleeding pattern. In contrast, they suggested that endoscopic clot removal and proper management with ET may be beneficial for patients with adherent clots (Forrest IIb). However, there are no associated data regarding bleeding GC.

Furthermore, recent studies have shown the utility of a topical spray using hemostatic agents for managing acute upper GI hemorrhage [12,15,16]. Kim et al. [12] showed that initial hemostasis was successfully achieved using hemostatic powder (EndoClot polysaccharide hemostatic system) with or without conventional hemostatic modalities in all 12 patients with Forrest 1b bleeding from gastric malignancies. In addition, rebleeding occurred in only two cases (16%) without any adverse events, a rate that was lower than that obtained with conventional hemostatic modalities (28–41%) [4,10,11]. Larger studies are expected to confirm the utility of this approach in hemostasis for GC bleeding.

Several studies have also shown the utility of an over-the-scope clip (OTSC) for patients with active NVUGIH bleeding [17,18,19]. However, we found no evidence of the utility of this approach for patients with GC bleeding.

### 2.4. Complications of ET

Only a few adverse events associated with ET for peptic ulcer bleeding have been reported, including induction of bleeding (1.7%) and perforation (0.6%) even when using combination therapy [20]. For bleeding GC, Kim et al. [12] used hemostatic powder in afflicted patients and noted no ET-related adverse events. While none of the four other studies using ET for patients with GC bleeding made any mention of ET-related adverse events, there were at least either no ET-related mortalities or no cases of operation resulting from adverse events such as perforation [4,9,10,11]. However, cases of hemorrhage induced by ET application might have been included among patients with hemostatic failure.

## 3. Transcatheter Embolotherapy for Bleeding from Unresectable GC

The efficacy of transcatheter angiographic embolization (TAE) in treating NVUGIH when medical or ET are insufficient has been shown in large studies [21,22,23]. In recent clinical studies, reporting the outcomes of ET for GC bleeding [4,9,10,11], TAE was performed for patients who failed ET and developed rebleeding after initial ET, and hemostasis was achieved in 83–100% of cases, except in 1 study with a low hemostatic rate of 33% [4]. However, in 3 out of 4 of these studies, only a few patients underwent TAE, and there was no detailed description of TAE [4,10,11]. In this section, literature searches were conducted in PubMed databases published in English for the period since 2013 using related key words, such as ‘GC’, ‘bleeding’ and ‘transcatheter’ (Figure 3). We also included one study which were referred in the ET sections with a detailed description of TAE for GC bleeding [9]. As a result, we reviewed three retrospective studies that did include a detailed description of TAE for GC bleeding, with our findings noted below (Table 2) [9,24,25].

### 3.1. Outcomes of TAE

The high technical success rates of hemostasis by TAE in patients with GC bleeding were reported to range from 85–100% [9,24,25]. However, the clinical success rate was lower ranging from 40–65% in two studies [24,25]. Although it seemed to be inferior to the successful initial hemostatic rate in patients who underwent ET for GC bleeding, there are little comparability between studies because the majority of patients who received TAE were unsuitable for or resistant to ET. This result was also inferior to the successful initial control rate of bleeding after TAE for NVUGIH (80–98%) [26]. Tumor-related bleeding limited the clinical success due to the direct invasion of vascular structures and development of chemotherapy- or RT-induced mucositis [26]. Three studies noted that the rebleeding rate after initial hemostasis ranged from 41% to 66% [9,24,25]. The median OS was 0.6–2.8 months, the median duration of response was 0.9–3.7 months, and the 30-day mortality rate was 25–60%, results that seemed worse than those in patients who underwent ET. However, these results cannot be comparable because survival time may depend more on tumor load, organ functions and performance status than on hemostatic procedure. The limited availability of treatment modalities for failure or rebleeding after both ET and TAE may influence the poor prognosis associated with TAE. One study found that active bleeding (*p* = 0.044) and a high transfusion requirement (*p* = 0.039) were associated with TAE failure [25]. In addition, successful TAE was shown to predict an improved 30-day survival after TAE according to univariate and multivariate analyses (*p* = 0.018 and *p* = 0.022; odds ratio, 0.132).

### 3.2. Indications of TAE

All patients underwent TAE for bleeding GC after ET failure in two studies [9,24]. In another study [25], an angiographic evaluation was performed as the initial evaluation because of the expected technical challenges based on the tumor extent and location on computed tomography (CT) or previous endoscopic features or because of the patients’ hemodynamic instability in 20 of 43 patients. Another study that compared patients with successful endoscopic hemostasis to those with unsuccessful endoscopic hemostasis followed by TAE concluded that TAE could be recommended in cases with large bleeding lesions (>2 cm) and non-exposed vessels bleeding with a tumor [9].

### 3.3. TAE Procedure

Two of the abovementioned studies described the TAE procedure in detail [24,25]. The most common angiographic abnormality was tumor staining without active extravasation (in 62% and 90%). Active bleeding, such as extravasation and pseudoaneurysm, was noted in 22% and 10% of patients (Figure 4). The most commonly embolized artery was the left gastric artery alone (42% and 70%); the left gastric artery combined with other vessels, such as the right gastroepiploic artery, gastroduodenal artery, right gastric artery, left hepatic artery, common hepatic artery, bilateral gastroepiploic artery, or right gastroepiploic artery plus the splenic artery, received embolization in 37% and 30% of patients. Embolization with particulate alone, such as gelatin sponge, polyvinyl alcohol, N-butyl cyanoacrylate, or tris-acryl gelatin microsphere, was performed in 72% and 90% of patients. In contrast, combined particulate and microcoil embolization was performed in 22% and 10% of patients.

### 3.4. Complications of TAE

In the management of NVUGIH [26], potential complications of TAE include contrast nephropathy, vascular injury at the puncture site or access vessels, nontarget embolization, and bowel ischemia. The overall reported adverse events rate is 6% to 9%, while that of major adverse events is <2%. However, among studies in which TAE was performed for patients with GC bleeding, one reported two cases of splenic infarction after TAE that were detected on CT as minor complications [25]. Another study reported no complications [24]. Procedure-related death was noted in all three studies.

## 4. Palliative RT for Bleeding from Unresectable GC

Palliative RT has been shown to be effective in patients with tumor bleeding from not only GC [27,28,29,30,31,32,33,34] but also malignancies of various origins [35,36,37,38]. However, only a few patients with initial hemostatic failure of ET or rebleeding after ET underwent hemostatic RT [4,9,11], and hemostatic RT is not common in most facilities in Japan [39]. The reason for such limited usage of this procedure may be because studies regarding palliative RT for bleeding GC are limited, most are retrospective, and conditions such as patient background characteristics, radiation dose, and assessment of hemostasis among the available studies. Therefore, the optimal dose fractionation for palliative RT for bleeding GC has not been established although a systematic review and meta-analysis did reveal its clinical benefit [34].

Literature searches were conducted in PubMed databases published in English for the period since 2013 using related key words, such as ‘GC’, ‘bleeding’ and ‘RT’ (Figure 5). We identified seven recent studies that reported the outcomes of palliative RT for GC bleeding [27,28,29,30,31,32,33] (Table 3 and Table 4) and a systemic review article [34] published. One study was a prospective phase II study [33], and six were retrospective [27,28,29,30,31,32]. Three of these studies included the assessment of not only bleeding but also pain and/or obstruction [28,32,33].

### 4.1. Outcomes of Palliative RT

Tey et al. [33] first revealed the efficacy of palliative RT on bleeding GC by the prospective approach in 2019, reporting a success rate of 80% with 36 Gy in 12 daily fractions (the biologically effective dose (BED) of 48.6 Gy). They also showed that palliative RT was well tolerated and resulted in an improvement in patients’ quality of life (QOL). On reviewing all 7 studies, including retrospective studies, the rate of successful hemostasis by palliative RT in all patients with GC bleeding ranged from 50% to 88% (Figure 6), although various dose fractionation regimens were attempted in each study [27,28,29,30,31,32,33]. Among them, the hemostatic rate in patients without concurrent chemotherapy was 50–80% [27,28,29,33]. The median OS was 2.1–5.3 months, and the median duration of response was 0.9–3.7 months, although the survival time was likely influenced by various factors, such as the time from the initial treatment, previous chemotherapy, the performance status (PS), and the existence of the metastasis [40]. In the prospective study, both univariable and multivariable analyses showed that the risk of death was higher for patients who did not respond to RT for GC bleeding than for responders, (hazard ratio 0.20 [95% confidence interval 0.07–0.57], *p* < 0.01 [33]). Another retrospective study showed that the median survival was significantly longer in patients who responded to palliative RT than in those who did not (113.5 vs. 47 days, *p* < 0.001) [28]. Two studies described the median time to hemostasis as two and 15 days (range, 1–9 and 1–84 days, respectively) [29,31].

Several studies investigated the factors influencing the treatment outcome. In the prospective study, both univariable and multivariable analyses showed that the age, gender, PS, TNM stage, and timing of chemotherapy were not associated with bleeding response [33]. Hiramoto et al. showed that the antrum as the primary site (*p* = 0.063) and peritoneal metastasis (*p* = 0.054) were more frequent in non-responders than in responders according to a univariate analysis [32]. Lee et al. showed that the absence of distant metastasis and the use of concurrent chemotherapy resulted in a better palliative RT response, and a biologically effective dose (using α/β = 10, BED_10_) ≥ 36 Gy was the most significant factor associated with the palliative RT response (*p* = 0.001) in the multivariate analysis [31]. We will discuss the correlation between the BED_10_ and bleeding response in detail later.

### 4.2. Condition of RT

Five of the seven studies mainly evaluated patients using CT [28,29,31,32,33], and two made no mention [27,30]. The gross tumor volume was defined as the whole stomach in one study [32], mixed whole and partial stomach in three studies [28,31,33], only the primary lesion in two studies [29,30], and not recorded in one study [27]. The field arrangement was the anterior-posterior field with parallel opposed lateral fields in three studies [27,29,33], opposed anterior-posterior field in two studies [28,30], and mixed in two studies [31,32].

Although the dose fractionation regimens varied among studies, the fraction size ranged from 2–8 Gy and the total dose from 6–60 Gy, corresponding to a BED of 7.2–50.8 Gy_10_. Chemoradiotherapy was administered to 9.2% (29/315) of patients. The correlation between the dose and response was controversial, and the optimal regimen remains unclear. Tey et al. [33] selected 36 Gy in 12 daily fractions in their prospective study, as Anderson had previously suggested that low-BED regimens (<41 Gy) predicted poorer local control compared than higher-BED regimens [41]. Lee et al. [31] showed that the radiation dose (BED_10_ ≥36 Gy) was significantly associated with bleeding cessation. However, several recent studies have suggested the usefulness of low-dose, short-course palliative RT because bleeding control was not affected by the total radiation dose [27,28,30], and there was no marked difference in the bleeding response rate between regimens with high BED (≥39 Gy) and low BED (<39 Gy) (*p* value = 0.39) according to a systemic review and meta-analysis [33]. However, the rebleeding rate in regimens with a low BED of <39 Gy [28,30] was shown to be higher (36–60%) than that in regimens with a high BED of ≥39 Gy (5–25%) [28,32,33]. These results suggest that low doses of irradiation may have a shorter hemostatic effect than higher doses, although even low doses of irradiation can lead to acute vascular damage to the tumor vessels. However, one study using a regimen with low-dose RT [30] showed that less than half of the patients required repeat RT despite rebleeding. They suggested that the durable effect of RT might prevent massive bleeding from the tumor; thus, slowly progressive anemia caused by slight bleeding might be controlled by appropriate blood transfusion.

It is important to evaluate promising regimens with low BED prospectively and to attempt to identify the optimal treatment regimen. Therefore, future studies should identify important factors, and prospective single-arm studies using regimens with low BED should be conducted in order to demonstrate the effectiveness of low-dose, palliative RT.

### 4.3. Toxicity

Six out of seven studies reported treatment-related toxicities (Table 4) [28,29,30,31,32,33]. Validated grading scales of the Common Terminology Criteria (CTC) were used. Grade ≥3 acute adverse events, such as vomiting, gastritis, GI obstruction, anorexia, neutropenia, and leukocytopenia, occurred in 0–20% of patients. All four patients suffered from neutropenia and leukocytopenia had received concurrent chemotherapy [29,30]. All patients with adverse events except for one with GI obstruction had their condition improved by conservative treatment. Two studies reported no Grade ≥3 adverse events. One study using a regimen with a high BED (48.6 Gy_10_) [33] suggested that treatment with three-dimensional conformal RT with anterior-posterior and lateral fields might contribute to a low rate of treatment toxicity (4%). Thus, the development and improvement of both tangible and intangible factors in RT should provide not only more effective but also safer treatment for patients in palliative situations.

### 4.4. The Assessment of the QOL

The prognosis in patients with unresectable bleeding GC was only 2–6 months despite the availability of various treatment modalities. Indeed, 4–66% of patients with bleeding GC who received palliative RT had a poor PS (≥3) [28,29,30,31,32,33]. However, while a QOL assessment should be routinely included in studies of palliative intervention, only two assessed the change in the QOL after palliative RT [30,33]. One prospective study [33] reported improvements in the fatigue, nausea/vomiting, and pain subscales of the EORTC Quality of Life Questionnaire C30 (EORTC QLQ-C30) in 50%, 28%, and 44% of patients at the end of RT and in 63%, 31% and 50% of patients at one month after RT, respectively. Furthermore, improvements in the dysphasia/pain subscales of the gastric-specific module (STO22) were seen in 42% and 28% of patients at the end of RT and in 44% and 19% of patients at 1 month after RT, respectively.

A retrospective study [30] using the low-dose, short-course regimen found that the PS had improved in six patients (25%) and that dietary intake became possible two weeks after RT in 2 of the 5 patients who had not been able to eat before RT. In addition, the median hospitalization period was 17 days, which was estimated to be shorter than the hospitalization period in patients who underwent RT using regimens with high BED, and some patients were able to be treated as out-patients. Palliative RT, especially short-course regimens, can help enhance patients’ QOL by relieving various painful symptoms, such as fatigue, nausea/vomiting, pain, and appetite loss, and allowing them to spend their precious, limited time at home in a calm environment, which appears to be a major concern for patients with a limited life expectancy [42,43].

The optimal dose of palliative RT considering not only the efficacy but also the patients’ status, including their general condition, prognosis, and QOL, remains to be determined.

## 5. Treatment Strategy for Bleeding from Unresectable Gastric Cancer

One major difference between the management of unresectable bleeding GC and that of benign bleeding lesions is that another treatment modality (palliative RT) besides ET and TAE can be applied in the clinical setting. However, we found either no comparative studies for these treatment modalities or no specific guidelines concerning the management of unresectable GC with bleeding including palliative RT. We will, therefore, now discuss treatment strategies for bleeding from unresectable GC, with our conclusions summarized in Figure 7.

When early upper GI endoscopy following hemodynamic resuscitation demonstrates that the bleeding source is GC, we should select the treatment option according to the Forrest classification of endoscopic stigmata. First, ET is recommended in cases of active bleeding (Forrest Ia and Ib) because a high rate of initial hemostasis can be expected with various ET modalities, which will help stabilize the patient’s general condition. If endoscopic hemostasis fails in patients with hemodynamic instability due to massive bleeding, emergent TAE should be considered in order to avoid a life-threating condition due to massive bleeding. In cases of hemostatic failure in patients without hemostatic instability, early palliative RT with conservative therapy, including fasting, PPI infusion and blood transfusion, should be considered, as an early hemostatic effect within a few days after the initiation of palliative RT can be expected [27,29,30].

Early palliative RT should be considered not only for patients with hemostatic failure but also for those with successful hemostasis and with inactive hemorrhage (Forrest IIa, IIb, IIc, and III), as a longer duration of response with few complications can be expected in palliative RT. In particular, regimens with a high BED of ≥39 Gy RT are preferable for preventing rebleeding as long as patients’ general condition and estimated prognosis allows such an approach.

## 6. Conclusions

This review suggests that bleeding from unresectable GC can be controlled safely in a majority of patients using hemostatic modalities, such as ET, TAE, and palliative RT. Further prospective, comparative studies for these hemostatic modalities considering not only the hemostatic efficacy of these modalities but also patients’ QOL are needed in order to establish treatment strategies for bleeding from unresectable GC.

## Figures and Tables

**Figure 1 biomedicines-07-00054-f001:**
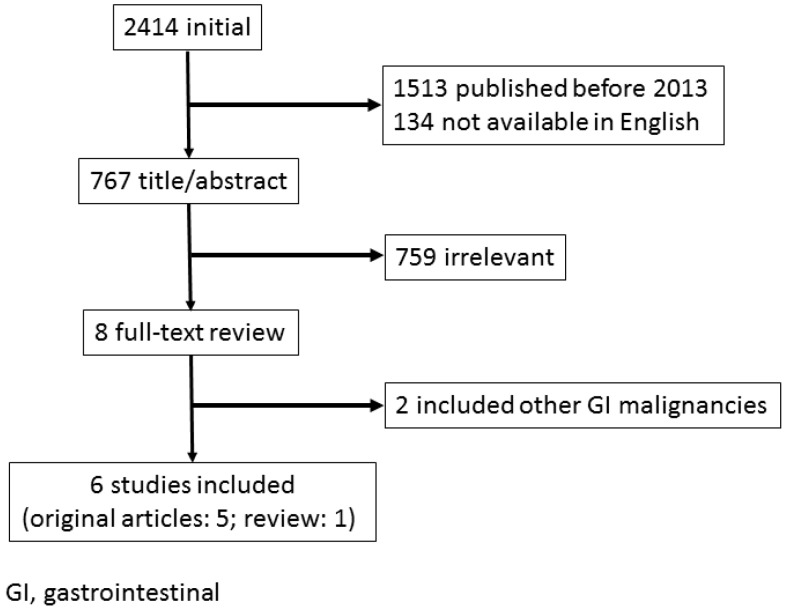
Study flow chart in endoscopic management for bleeding gastric cancer.

**Figure 2 biomedicines-07-00054-f002:**
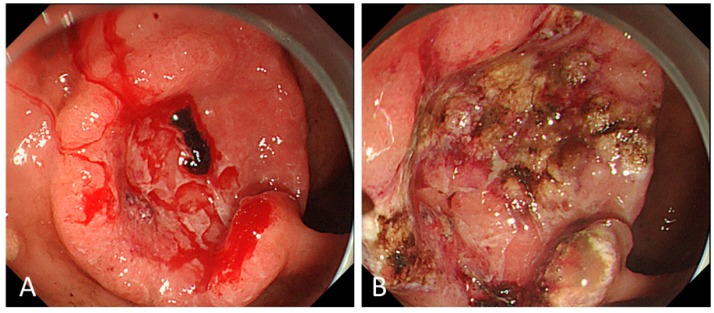
Endoscopic therapy for gastric cancer bleeding. Argon plasma coagulation (APC) was performed for oozing hemorrhage (Forrest Ib) from the ulcerative tumor at the anterior wall of the antrum (**A**) and hemostasis was successfully achieved (**B**).

**Figure 3 biomedicines-07-00054-f003:**
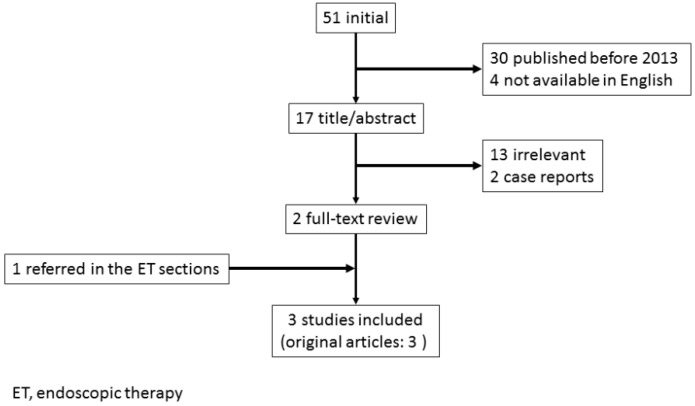
Study flow chart in transcatheter angiographic embolization for bleeding gastric cancer.

**Figure 4 biomedicines-07-00054-f004:**
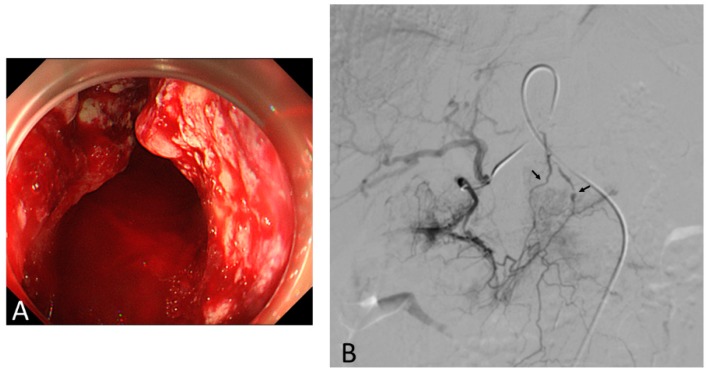
A large ulcerative tumor with spurting hemorrhage from the cardia to the whole body was confirmed endoscopically (**A**); however, endoscopic hemostasis failed due to massive bleeding. A selective right gastric artery angiogram revealed two contrast extravasations (arrows), one of which accompanied pseudoaneurysm (**B**). However, transcatheter angiographic embolization was not indicated because of the failure of super-selective catheterization.

**Figure 5 biomedicines-07-00054-f005:**
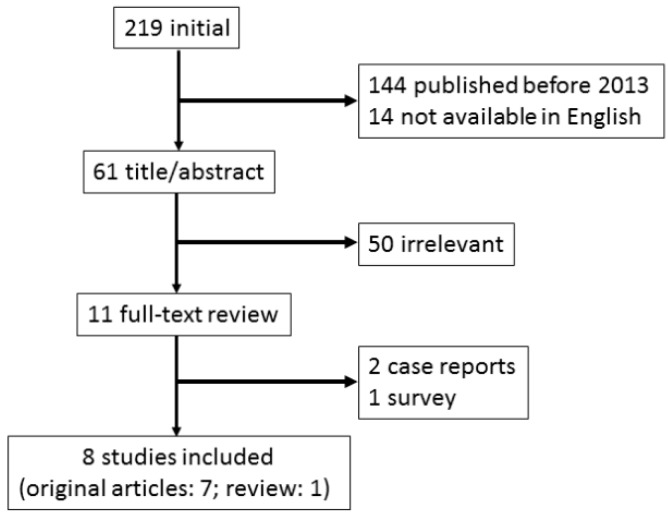
Study flow chart in palliative radiotherapy for bleeding gastric cancer.

**Figure 6 biomedicines-07-00054-f006:**
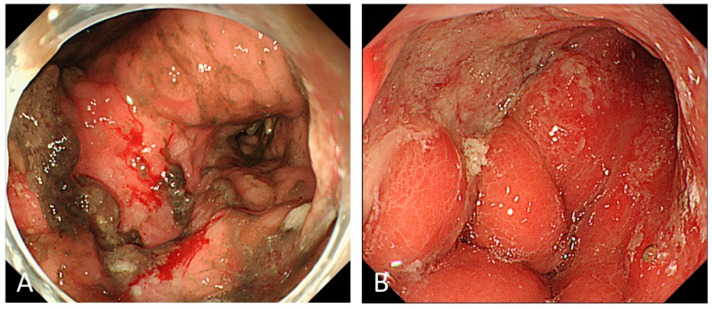
Palliative radiotherapy (PRT) for bleeding gastric cancer. PRT with 30 Gy in 10 fractions was applied to an easy-bleeding tumor at the anterior wall of angle (**A**). No bleeding event or anemia progression had been experienced by four months after starting PRT and ulceration of the tumor was confirmed endoscopically (**B**).

**Figure 7 biomedicines-07-00054-f007:**
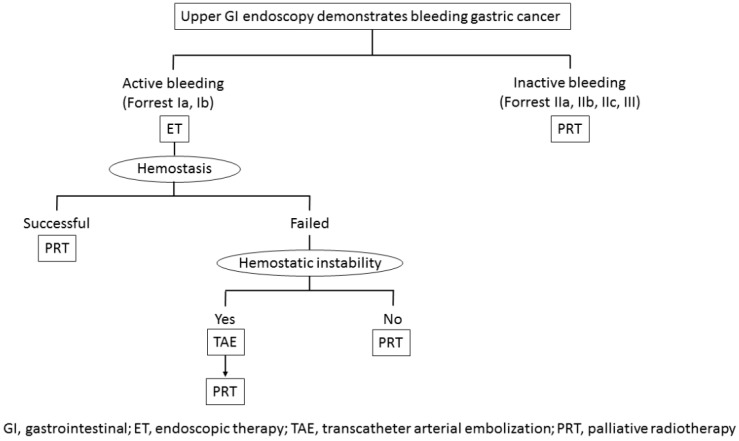
Treatment strategy for bleeding from unresectable gastric cancer.

**Table 1 biomedicines-07-00054-t001:** Characteristics and results of endoscopic therapy for bleeding from gastric cancer.

Author/Year	Patients				Modalities	Successful Hemostasis, *n* (%)	Rebleeding, *n* (%)	Prognosis	
	Number	Stage, *n* (%)	Previous Treatment	Forrest Class I a/b, *n* (%)				Median OS (Months)	30-Day Mortality Rate
Koh KH 2013 [9]	45	NR	NR	20/25 (100%)	Thermal therapy, clipping, injection, spray	14 (31%)	NR	NR	18.8%
Kim YI 2013 [10]	113	110 (97.3%)	CT; 22	17/59 (67%)	Thermal therapy, clipping, injection, spray	105 (92.9%)	43 (41%)	3.2	15.9%
Song IJ 2017 [11]	106	105 (99.1%)	CT; 53, RT; 1, CRT; 10	6/59 (61%)	Thermal therapy, clipping, injection	88 (83.0%)	30 (28.3%)	3.9 in non-rebleeding patients 2.7 in rebleeding patients	22.6%
Park H 2017 [4]	64	41 (64.1%)	CT; 52, RT; 3	NR	Thermal therapy, clipping, injection, spray	47 (73.4%)	17 (36.2%)	6.5	NR
Kim YJ 2018 [12]	12*	NR	CT; 6	0/12 (100%)	Hemostatic powder application 5 required combined therapy (thermal therapy, clipping, injection)	12 (100%)	2 (16%)	NR	0%

NR, not recorded; OS, overall survival; CT, chemotherapy; RT, radiotherapy; CRT, chemoradiotherapy; * Included a gastrointestinal stromal tumor.

**Table 2 biomedicines-07-00054-t002:** Characteristics and results of transcatheter angiographic embolization for bleeding from gastric cancer.

Author/Year	Patients					Successful Hemostasis, n (%)		Rebleeding, *n* (%)	Prognosis		Complications
	Number	Stage, *n* (%)	Endoscopic Evaluation	Forrest Class I a/b, *n* (%)	ET *n* (%)	Technical	Clinical		Median OS (Months)	30-Day Mortality Rate	
Koh KH 2013 [9]	31	NR	31	10/21 (100%)	31 (100%)	31 (100%)	31/31 (100%)	5 (16%)	NR	25.8%	NR
Meehan T 2014 [24]	10	NR	10	NR	10 (100%)	10/10 (100%)	4/10 (40%)	43 (41%)	0.6	60.0%	0
Park S 2017 [25]	43	28 (68.3%)	23	4/16 (86%)	9 (20.9%)	34/40 (85%)	26/40 (65%)	7 (26%)	2.8	25.0%	Splenic infarction; 2

NR, not recorded; OS, overall survival; CT, chemotherapy; ET, endoscopic therapy; SIR, Society of Interventional Radiology.

**Table 3 biomedicines-07-00054-t003:** Characteristics and results of palliative radiotherapy for bleeding from gastric cancer.

Author/Year	Index Symptom	Patients			Radiotherapy		Chemotherapy			Successful Hemostasis, % (*n*)	Rebleeding % (*n*)	Survival (Months)	Duration (Months)	QOL Assessment
		Number	Stage	PS ≥ 3, *n* (%)	Dose/Fraction	BED (Gy_10_)	Previous	Concurrent	Additional					
Chaw CL 2014 [27]	Bleeding	52	Mixed	NR	8 Gy/1 fr or 20 Gy/5 fr	8.8 or 28	14	0	3	50% (22/44)	NR	5.3	NR	NR
Tey J 2014 [28]	Bleeding, pain, obstruction	115	Mixed	11 (9.5%)	8–40 Gy/1–16 fr	14.4–50	9	0	10	80.6% (83/103)	30% (25/83) BED ≤ 39 36% (17/47) BED > 39 22% (8/36)	2.8	3.3	NR
Kondoh C 2015 [29]	Bleeding	15	Metastatic	10 (66%)	Median 30 Gy (30–40 Gy/10–20 fr)	Median 39 (23–48)	9	5	5	73% (11/15- 7 in RT, 4 in CRT)	36% (4/11- 2 in RT, 2 in CRT)	2.1	0.9	NR
Kawabata H 2017 [30]	Bleeding	18	Mixed	4 (22%)	6 Gy/3 fr	7.2	13	2	8	55% (10/18)	60% (6/10)	NR	NR	PS, Dietary intake
Lee YH 2017 [31]	Bleeding	42	Mixed	8 (19%)	Median 39.6 Gy/20 fr (14–50.4 Gy/7–28 fr)	Median 47 (16.8–59.4)	31	7	NR	69% (29/42)	37% (11/29)	3.1	3.7	NR
Hiramoto S 2018 [32]	Bleeding, obstruction	23	Mixed	1 (4.3%)	Median 42 Gy/20 fr (30–60 Gy/10–30 fr)	Median 50.8 (39–72)	10	15	8	88.8% (16/18)	25% (4/16)	3.9	3.4	NR
Tey J 2019 [33]	Bleeding, pain, obstruction	50	Mixed	5 (10%)	36 Gy/12 fr	48.6	5	0	7	80% (40/50)	5% (2/40)	2.7	3.4	EORTCQLQ-C30

PS, Performance status; BED, biologically effective dose; QOL, quality of life; NR, not recorded; EORTCQLQ-C30, The European Organization for Research and Treatment of Cancer Quality of Life Questionnaire C30.

**Table 4 biomedicines-07-00054-t004:** Toxicity of palliative RT for bleeding from gastric cancer.

Author/Year	Toxicity, *n* (%)	Acute Toxicity (CTC)			Late Toxicity
		Gastrointestinal (Grade ≥ 3)	Skin/Connective Tissue (Grade ≥ 3)	Others (Grade ≥ 3)	
Chaw CL 2014 [27]	NR	NR	NR	NR	NR
Tey J 2014 [28]	3 (2%)	Vomitting; 1 Gastritis; 1	0	Anorexia; 1	0
Kondoh C 2015 [29]	3 (20%)	0	0	Neutropenia; 3 in CRT	0
Kawabata H 2017 [30]	2 (11%)	GI obstruction; 1	0	Leukocytopenia; 1 in CRT	0
Lee YH 2017 [31]	0	0	0	0	0
Hiramoto S 2018 [33]	0	0	0	0	0
Tey J 2019 [33]	2 (4%)	Gastritis; 1	0	Anorexia; 1	0

CTC, Common Terminology Criteria; NR, not recorded.

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
