# Peer review of "Management of Bleeding from Unresectable Gastric Cancer"

_biomedicines, 2019, doi:10.3390/biomedicines7030054_

Reviewer 1 Report

The following are my comment for Drs. Kawabata and Misuzu, on their paper on ‘Management of bleeding from unresectable gastric cancer’.

 Although clinical question was interesting, I have several concerns about interpretation of previous works, especially endoscopic hemostasis method. I believe my comments will help improve the paper and give the reader a better perspective on the relevance and importance of this report for patients with unresectable gastric cancer bleeding.

 1.     Describe how collect eligible papers. Authors should describe the method such as database name, duration, search key words in revised manuscript. E.g. PubMed, during 2009-2018, “gastric cancer bleeding” etc.

2.     Delete about ESGE guideline sentences. Your review article includes new previous finding data and not need to state old foreign country guideline.

3.     Authors should add sentences about complications about endoscopic therapy.

4.     I disagree your treatment strategy about endoscopic therapy. I agree usefulness of endoscopy for first diagnostic tool and hemostatic option for spurting bleeding. However, endoscopic therapy occurred high rebleeding rate and require repeat procedures. Please reconsider about usefulness of RT because RT did not require repeat procedure and had low complication rate.       

5.     Authors should add a Figure about recommended treatment strategy. In addition, I strongly recommend adding Figures about endoscopic image, angiography image. If authors did not have these figures, please get permission from referred papers.

6.     Authors should change the word “nil” to “0” in Table 2 and 4.

7.     Please confirm reference year “since 2014” in line 214.

Author Response

RESPONSE TO REVIEWER 1:

We wish to express our appreciation to the Reviewer 1 for his or her insightful comments, which have helped us significantly improve the paper.

 Comment 1: Describe how collect eligible papers. Authors should describe the method such as database name, duration, search key words in revised manuscript. E.g. PubMed, during 2009-2018, “gastric cancer bleeding” etc.

 Response:

In accordance with Reviewer 1’s comment, I described the method of database name, duration, and search key words in each section.

Comment 2Delete about ESGE guideline sentences. Your review article includes new previous finding data and not need to state old foreign country guideline.

 Response:

In accordance with Reviewer 1’s comment, I deleted ESGE guideline sentences in each section.

 Comment 3Authors should add sentences about complications about endoscopic therapy.

 Response:

In accordance with Reviewer 1’s comment, I added sentences about complications of endoscopic therapy as below.

 Kim et al. using hemostatic powder for patients with GC bleeding described there were no ET-related adverse events. Although other 4 studies using ET for patients with GC bleeding had no mention of ET-related adverse events, there were at least either no ET-related mortality or no cases of operation for adverse events such as perforation. However, induction of hemorrhage by applying ET might be included in patients with hemostatic failure.

 Comment 4: I disagree your treatment strategy about endoscopic therapy. I agree usefulness of endoscopy for first diagnostic tool and hemostatic option for spurting bleeding. However, endoscopic therapy occurred high rebleeding rate and require repeat procedures. Please reconsider about usefulness of RT because RT did not require repeat procedure and had low complication rate.

 Response:

I agree to your treatment strategies. In accordance with Reviewer 1’s comment, I revised the section of the treatment strategy as below.

In summary, first, ET is recommended in cases of active bleeding because a high rate of initial hemostasis can be expected with various ET modalities. If endoscopic hemostasis fails in patients with hemodynamic instability due to massive bleeding, emergent TAE is considered in order to avoid a life-threating condition. In cases of hemostatic failure in patients without hemostatic instability, early palliative RT should be considered, as an early hemostatic effect within a few days after the initiation of palliative RT can be expected.

Early palliative RT is also considered for those with successful hemostasis and with inactive hemorrhage, as a longer duration of response with few complications can be expected in palliative RT.

 Comment 5Authors should add a Figure about recommended treatment strategy. In addition, I strongly recommend adding Figures about endoscopic image, angiography image. If authors did not have these figures, please get permission from referred papers.

 Response:

In accordance with Reviewer 1’s comment, I added figures about endoscopic image and angiography image.

 Comment 6Authors should change the word “nil” to “0” in Table 2 and 4.

 Response:

In accordance with Reviewer 1’s comment, I changed the word “nil” to “0” in Table 2 and 4.

Comment 7 Please confirm reference year “since 2014” in line 214.

 Response:

In accordance with Reviewer 1’s comment, I corrected the reference year to “since 2014” in line 214.

Reviewer 2 Report

I agree with authors that management of bleeding from unresectable gastric cancer is important issue for patient care. However, there seems several points authors should reconsider. 

Authors reviewed and compared the efficacy reported by several studies on different methods of hemostasis, namely, endoscopic-, trans catheter- and radiation therapy. Authors stated endoscopic therapy is superior to the other two treatments (3.1., 5) despite there are little comparability between studies which authors cited. There is no study directly compare different method and there could be significant difference on patient background between studies. The conclusion should be reconsidered. 

In the part 2 on endoscopic management, authors seem emphasise ESGE guidelines too much, which is mainly applied for benign bleeding. Authors may focus more on management of malignant bleeding. 

Author Response

RESPONSE TO REVIEWER 2:

We wish to express our appreciation to the Reviewer 2 for his or her insightful comments, which have helped us significantly improve the paper.

 Comment: I agree with authors that management of bleeding from unresectable gastric cancer is important issue for patient care. However, there seems several points authors should reconsider. 

Authors reviewed and compared the efficacy reported by several studies on different methods of hemostasis, namely, endoscopic-, trans catheter- and radiation therapy. Authors stated endoscopic therapy is superior to the other two treatments (3.1., 5) despite there are little comparability between studies which authors cited. There is no study directly compare different method and there could be significant difference on patient background between studies. The conclusion should be reconsidered. 

In the part 2 on endoscopic management, authors seem emphasise ESGE guidelines too much, which is mainly applied for benign bleeding. Authors may focus more on management of malignant bleeding. 

 Response:

In accordance with Reviewer 2’s comment, I revised the sentence in conclusion as below:

 Further prospective, comparative studies among these hemostatic modalities considering not only the hemostatic efficacy of these modalities but also patients’ QOL are needed in order to establish treatment strategies for bleeding from unresectable GC.

 And, I deleted sentences of ESGE guideline and focused more on management of malignant bleeding.

 Comment 1: I'd like to see why these statements were made - is there evidence for them?: Low-salt soups optimizing the flavors of the raw ingredients without any chemical seasonings avoid irritating the GI tract and absorb smoothly, as shown from the patients’ comments. The soups may not only relieve their stress and provide a feeling of relief but also help improve their spiritual QOL due to the satisfaction and comfort derived from drinking the soup.

 Response: We thank the Reviewer 1 for this pertinent comment.

 In accordance with Reviewer 1’s comment, we have added the sentence with references which have suggested the harmful effects of chemical seasonings and food additives as below:

 Various types of chemical seasoning or food additives have been suggested to cause digestive symptoms (10, 11).

10. Sharma S.: Food preservatives and their harmful effects. Int J Sci Res 2015;5:1-2.

11. U.S. Department of Health and Human Services. U.S. food and drug administration. Questions and Answers on Monosodium glutamate (MSG). https://www.fda.gov/food/ingredientspackaginglabeling/foodadditivesingredients/ucm328728.htm (last accessed September 18, 2017).

 And, the second sentence was led from our consideration based on the patients’ comments and the related evidences shown in the introduction because we could find no definite evidences.

 Thank you again for your comments on our paper. I trust that the revised manuscript is suitable for publication.

 The following are my comment for Drs. Kawabata and Misuzu, on their paper on ‘Management of bleeding from unresectable gastric cancer’.

 Although clinical question was interesting, I have several concerns about interpretation of previous works, especially endoscopic hemostasis method. I believe my comments will help improve the paper and give the reader a better perspective on the relevance and importance of this report for patients with unresectable gastric cancer bleeding.

 1.     Describe how collect eligible papers. Authors should describe the method such as database name, duration, search key words in revised manuscript. E.g. PubMed, during 2009-2018, “gastric cancer bleeding” etc.

2.     Delete about ESGE guideline sentences. Your review article includes new previous finding data and not need to state old foreign country guideline.

3.     Authors should add sentences about complications about endoscopic therapy.

4.     I disagree your treatment strategy about endoscopic therapy. I agree usefulness of endoscopy for first diagnostic tool and hemostatic option for spurting bleeding. However, endoscopic therapy occurred high rebleeding rate and require repeat procedures. Please reconsider about usefulness of RT because RT did not require repeat procedure and had low complication rate.       

5.     Authors should add a Figure about recommended treatment strategy. In addition, I strongly recommend adding Figures about endoscopic image, angiography image. If authors did not have these figures, please get permission from referred papers.

6.     Authors should change the word “nil” to “0” in Table 2 and 4.

7.     Please confirm reference year “since 2014” in line 214.

I agree with authors that management of bleeding from unresectable gastric cancer is important issue for patient care. However, there seems several points authors should reconsider. 

Authors reviewed and compared the efficacy reported by several studies on different methods of hemostasis, namely, endoscopic-, trans catheter- and radiation therapy. Authors stated endoscopic therapy is superior to the other two treatments (3.1., 5) despite there are little comparability between studies which authors cited. There is no study directly compare different method and there could be significant difference on patient background between studies. The conclusion should be reconsidered. 

In the part 2 on endoscopic management, authors seem emphasise ESGE guidelines too much, which is mainly applied for benign bleeding. Authors may focus more on management of malignant bleeding.

Round  2

Reviewer 2 Report

As the reviewer 1 stated, authors should describe how to select the references. PubMed shows many other literatures with key words authors show. There may be selection bias of the literatures. Considering the results of reference #28, RT seems to have most evidence on haemostasis for gastric cancer bleeding. Please clarify the selection and exclusion criteria and the search results: how many literatures were found and how many was excluded by what reasons. Authors may add a figure of the selection tree. 

 Again, the description in 3.1. seems inappropriate. Authors compare haemostatic success rate between TAE and ET, and describe TAE was inferior to ET. However, there are little comparability between studies because the tumour of patients who received TAE must be unsuitable for or resistant to ET. Moreover, survival time may depend more on tumour load, organ functions and performance status than on haemostatic procedure.

Author Response

RESPONSE TO REVIEWER 2:

We wish to express our appreciation to the Reviewer 2 for his or her insightful comments, which have helped us significantly improve the paper.

 Comment: As the reviewer 1 stated, authors should describe how to select the references. PubMed shows many other literatures with key words authors show. There may be selection bias of the literatures. Considering the results of reference #28, RT seems to have most evidence on haemostasis for gastric cancer bleeding. Please clarify the selection and exclusion criteria and the search results: how many literatures were found and how many was excluded by what reasons. Authors may add a figure of the selection tree. 

 Again, the description in 3.1. seems inappropriate. Authors compare haemostatic success rate between TAE and ET, and describe TAE was inferior to ET. However, there are little comparability between studies because the tumour of patients who received TAE must be unsuitable for or resistant to ET. Moreover, survival time may depend more on tumour load, organ functions and performance status than on haemostatic procedure.

 Response:

First, in accordance with Reviewer 2’s comment, I clarified the selection and exclusion criteria by adding a figure of the selection tree in each section.

 Second, I agree with Reviewer 2’s comment on the outcomes of TAE. I revised the description to emphasize the incomparability between ET and TAE by referring the Reviewer 2’s comment.

 Thank you again for your comments on our paper. I trust that the revised manuscript is suitable for publication.